# Motor Competence Assessment (MCA) Scoring Method

**DOI:** 10.3390/children9111769

**Published:** 2022-11-17

**Authors:** Luis Paulo Rodrigues, Carlos Luz, Rita Cordovil, André Pombo, Vitor P. Lopes

**Affiliations:** 1Instituto Politécnico de Viana do Castelo, Escola Superior Desporto e Lazer de Melgaço, 4900-347 Viana do Castelo, Portugal; 2Research Center in Sports Sciences, Health and Human Development (CIDESD), 5000-801 Vila Real, Portugal; 3Research Center in Sports Performance, Recreation, Innovation and Technology (SPRINT), 4960-320 Melgaço, Portugal; 4Escola Superior de Educação de Lisboa, Instituto Politécnico de Lisboa, 1549-003 Lisboa, Portugal; 5Centro Interdisciplinar de Estudos Educacionais, 1549-003 Lisboa, Portugal; 6Faculdade de Motricidade Humana, Universidade de Lisboa, 1499-002 Cruz Quebrada-Dafundo, Portugal; 7CIPER, Faculdade de Motricidade Humana, Universidade de Lisboa, 1499-002 Cruz Quebrada-Dafundo, Portugal; 8Instituto Politécnico de Bragança, Campus de Santa Apolónia, 5300-223 Bragança, Portugal

**Keywords:** human development, motor development, motor test, motor performance, lifespan

## Abstract

The Motor Competence Assessment (MCA) is a quantitative test battery that assesses motor competence across the whole lifespan. It is composed of three sub-scales: locomotor, stability, and manipulative, each of them assessed by two different objectively measured tests. The MCA construct validity for children and adolescents, having normative values from 3 to 23 years of age, and the configural invariance between age groups, were recently established. The aim of this study is to expand the MCA’s development and validation by defining the best and leanest method to score and classify MCA sub-scales and total score. One thousand participants from 3 to 22 years of age, randomly selected from the Portuguese database on MC, participated in the study. Three different procedures to calculate the sub-scales and total MCA values were tested according to alternative models. Results were compared to the reference method, and Intraclass Correlation Coefficient, Cronbach’s Alpha, and Bland–Altman statistics were used to describe agreement between the three methods. The analysis showed no substantial differences between the three methods. Reliability values were perfect (0.999 to 1.000) for all models, implying that all the methods were able to classify everyone in the same way. We recommend implementing the most economic and efficient algorithm, i.e., the configural model algorithm, averaging the percentile scores of the two tests to assess each MCA sub-scale and total scores.

## 1. Introduction

Motor competence (MC) relates to human movement, its development, and performance. It has been described as the ability to be skilful in a wide variety of gross motor skills (stability, locomotor, and manipulative) that are associated with multiple developmental outcomes including physical health [1,2,3], psychological, social-emotional, and cognition/achievement [4,5,6,7]. Importantly, it is expected that adequate levels of MC will facilitate the proficiency of novel motor tasks throughout the lifespan and the learning of new skills [8]. 

The Motor Competence Assessment (MCA) is an instrument developed to assess motor competence across a lifespan. For the first time, it is possible to assess MC from childhood to old age using the same instrument, without a developmental (age) ceiling effect and using a feasible and objective test battery [8,9,10]. Although there are other instruments specifically developed or adapted to evaluate motor competence throughout a lifespan, such as the Test of Motor Competence (TMC) [11] and the last form of the Körperkoordinationstest für Kinder (KTK3+) [12], the first includes a fine motor component that is outside our definition of MC, and the second shows a clear ceiling effect on one of the tests used (balancing backwards) and no locomotor component is included.

The MCA is a quantitative test battery composed of six tests divided into three sub-scales: stability, locomotor, and manipulative. Each sub-scale is calculated from two tests, objectively measured. All motor tests in the MCA are quantitative (product-oriented), with no developmental (age) ceiling effect, and are easy to execute even with little practice. After determining the construct validity of the MCA for children and adolescents [9], the normative values from 3 to 23 years of age were established [10]. Very recently, the MCA configural invariance between age groups was tested, proving the usefulness of the MCA model throughout growth and development periods from early childhood to young adulthood [12].

Typically, the scales or batteries used to assess motor development or motor competence of children use a classification system that involves transforming the tests’ raw data to standard scores relative to age and sex, but with a limited range (from 1 to 10, or more usually 1 to 20). Examples are the KTK, the Peabody Developmental Motor Scales 2 (PDMS-2) [13], the Test of Gross Motor Development 3 (TGMD-3) [14], and the Movement Assessment Battery for Children (MABC-2) [15], where a singular value of a standard score can accommodate a range of results from the test. For example, at the age of six, all values between 15 and 23 s on the one-leg balance test of the MABC are transformed into a standard score of eight. The same reduction in information happens when assessing the sub-scales or components, and the total value, or motor quotient, of the child. In such a procedure, a chunk of information is lost each time a transformation to standard scores is performed, and consequently we lose possible discrimination between children’s real values. Since the MCA always uses quantitative ratio scales to assess the six tests, we argue all this information relative to age and sex (normative percentile values) should be used to better discriminate between subjects.

Although the MCA has been used with different populations and cultures [16,17,18], the method for scoring each of the three sub-scales (locomotor, stability, and manipulative) and for total MCA score, still needs to be defined. The initial theoretical framework on the development of motor skills [19,20] proposed an equal participation of the locomotor, stability, and manipulative components in overall motor competence. However, this structural relationship needs to be tested relative to the circumstances of the tests used to mark each component, and to the developmental ages where the evaluation takes place. Furthermore, since each MCA test is not a perfect marker of the component but only a proxy of it, there is a need to interpret possible different weights of the tests in the classification method of the components (sub-scales) and total MCA scores. The previous work undertaken on the validation of the MCA established the metrics for these questions, i.e., the relative weight of each test to the sub-scale according to different age periods [8,9,10]. Hence, the aim of this study will be to expand the MCA’s development and validation by using the previous findings on the developmental characteristics of the MCA tests to define the best and leanest method to score and classify MCA sub-scales and total score.

## 2. Materials and Methods

### 2.1. Participants

An initial pool of 2150 participants (1003 females; 1147 males) from the Portuguese database on MC, a convenience sample consisting of students from preschool to university, was utilized to select the sample for this study. A computer program was used to randomly select 1000 participants (250 per age group from 3 to 6, 7 to 10, 11 to 16, and 17 to 23 years of age, gender-balanced within age groups). No differences were found on any of the MCA tests (all p’s > 0.50), between the selected and the non-selected participants. All participants showed no motor or cognitive impairment.

The study was approved by the Ethics Committee of the Faculty of Human Kinetics at the University of Lisbon and the Scientific Council of the School of Sports and Leisure at the Polytechnic Institute of Viana do Castelo. All school directors authorized the study, and the informed consent of adult participants or parents/tutors of underage children was obtained. All children gave their verbal assent prior to data collection. All procedures were in accordance with the 1964 Declaration of Helsinki and its later amendments [21].

### 2.2. Instruments and Procedures

The MCA is composed of three sub-scales, stability, locomotor, and manipulative. Each sub-scale has two tests, specifically: lateral jumps (LJs) and shifting platforms (SPs) for stability; standing long jump (SLJ) and 4 × 10 m shuttle run (SHR) for locomotor; and ball kicking velocity (BKV) and ball throwing velocity (BTV) for the manipulative sub-scale. All tests were assessed in a quantitative scale (i.e., distance, time, number of executions, or velocity), and do not have a ceiling effect related to age or sex (for full description see Rodrigues et al., 2019). The literature describes the tests’ reliability as ranging from good to excellent [22,23,24,25], and the values of the Intraclass Correlation Coefficient (ICC)were 0.95, 0.99, 0.97, 0.99, 0.98, and 0.98 respectively for SP, JS, SHR, SLJ, BKV, and BTV.

Participants completed a 10 min general and standardized warm-up before the testing. The test setting was organized in small groups (5 participants per task). Trained examiners administered the tests. A proficient demonstration and a verbal explanation of each test were provided before each test. A test trial was always provided to the participant before the actual test administration. Instructions were given for participants to perform each task at their maximum. Only motivational feedback was given during the test. All data collection was supervised by one of the authors of this study. Testing always took place in a gymnasium.

For the calculation of the sub-scales and total MCA, participants’ results in each of the MCA tests were transformed into age- and sex-normative values (percentiles) according to the process previously described [8]. Three different procedures to calculate the sub-scales values were tested according to three alternative models [25]: (1) the Weighted Age Group Model (WAGM) accounted for different weights (percentage) of each test’s representation in the final sub-scale score, according to the age groups’ previous results [8]; (2) the Overall Model (OM)used the same weight of each test, independent of the age, for calculating the sub-scale score, according to the loading coefficient of the model found for the overall sample [8]; and (3) the Configural Model (CM), in which all tests represented an equal value for the sub-scale’s calculation (see Table 1).

According to the different methods used, the formula to calculate each sub-scale score was:|((LC test 1/(LC test 1+ LC test 2)) * P test 1) + ((LC test 2/(LC test 1+ LC teste 2)) * P test 2)|(1)
where LC = Loading Coefficient and P = percentile value.

After defining the best method for calculating the sub-scales scores, identical methodological procedures were followed to calculate the total MCA score. The relative representation of each sub-scale was used for calculating the final MCA score according to the same three models previously used (WAGM, OM, and CM):**TOTAL MCA =** (((LC1+LC2)/sum LC) * STAB) + (((LC3+LC4)/sum LC) * LOC) + (((LC5+LC6)/sum LC) * MAN)(2)
where LC1 = Loading Coefficient test 1; STAB = stability sub-scale score; LOC = locomotor sub-scale score; MAN = manipulative sub-scale score.

### 2.3. Statistical Analysis

The results for the sub-scales and total MCA according to the three tested models were compared with the values produced by the Weighted Age Group Model (WAGM), the one selected as our reference method. This choice relates to the fact that the WAGM model provides more information on the weight of each test to the sub-scale score and the total score (configural, loading coefficients, and age-related information). The Intraclass Correlation Coefficient tested for the relationship between the results accounting for the absolute value; the Cronbach’s Alpha was used as a measure of reliability. The Bland–Altman plot with limits of agreement was used to describe agreement between the three methods taking the WAGM as the reference method [26,27].

Subsequently, the chosen method for calculating the MCA sub-scales scores was then used with our data to find each sub-scale score, and the MCA total score was then assessed according to the three different models previously used (WAGM, OM, and CM). 

Initial analyses were made using IBM SPSS Statistics for Windows, Version 26.0. (Armonk, NY, USA: IBM Corp). The Bland–Altman plot and subsequent analyses were calculated in MedCalc^®^ Statistical Software version 20.106 (MedCalc Software Ltd., Ostend, Belgium; https://www.medcalc.org; accessed on 1 February 2022).

## 3. Results

As depicted in Table 2, an almost perfect reliability level was found between the tested methods when compared to the reference method, with values ranging from 0.999 to 1.000 in the ICC and the Cronbach’s Alpha. Additionally, the mean difference (bias) resulting from the Bland–Altman technique always returned a very low value, very close to zero (not significantly different from zero in all cases except for WAGM * CM), with also a very narrow interval between 95% limits of agreement (Figure 1).

The sub-scales scores according to the CM method were then used (Table 3) with our data to find the MCA total score according to the three different models previously used (WAGM, OM, and CM).

When looking at Figure 1, which shows the Bland–Altman plot with the 95% limits of agreement, all differences between the tested (predicted) models and the reference values found when the WAGM model was used, were very small. Even the biggest differences were within a 0.1 percentile margin of error, meaning that any difference between the three tested models was negligible. Very few cases were beyond the 95% limits of agreement.

## 4. Discussion

The aim of this study was to determine the best algorithm for assessing the MCA sub-scales and total scores, and one representative of the developmental MC relationships. All test batteries for motor competence or motor development use sub-scales and total scores. In fact, the MCA has been used in recent years with each author using a composite of the tests to represent sub-scales and total MCA scores e.g., [16,17,18]. This practice has reflected the theoretical model assumed for MCA, long used in the field, that all tests and sub-scales have equal participation in the determination of the motor competence, but this had not been tested until now. To test this, we used three different models previously fitted to a sample of 1000 participants within 3 to 23 years of age, and their loading weights were used for calculation of the sub-scales and total MCA [26]. Since we hypothesized that the better-adjusted model should consider age-related constraints, the Weighted Age Group Model values were used as a reference model to compare to the other two: an Overall Model (using all the sample), and the Configural Model where all weights are assumed to be equal.

The analysis showed that no substantial differences could be found between the three methods used for assessing the sub-scales. The reliability values were perfect or in the vicinity of perfection (0.999 to 1.000), which implies that all the methods were able to classify everyone in the same way. The Bland–Altman plots represented in Figure 1 show the dimension of the differences between the reference model and the other two. The 95% limits of agreement found for both the CM and the OM resulted in a very narrow range, between −0.01 and 0.01 percentiles. That means that using either of the three methods would produce almost the same results for all individuals tested. Furthermore, the Bland–Altman bias analysis showed negligible values, indicating that all the methods can be used with similar results independent of the test scores’ magnitude.

Given these results for the sub-scales, we suggest implementing the most economic and efficient algorithm, i.e., the configural model algorithm, averaging the percentile scores of the two tests to assess each sub-scale. This method was used with our data to find each sub-scale score, and the MCA total score was then assessed according to the three different models previously used (WAGM, OM, and CM). Again, the results proved very similar in outcome, independent of the method used, with reliability values ranging from 0.999 to 1.000, and the 95% confidence intervals for the differences found for both the CM (−0.0010 to 0.000) and the OM (−0.0011 to −0.0001) were very small.

Other scoring methods used in comparable test batteries usually include the transformation of individual test’s raw values into z-scores related to age and sex (test z-score), and the transformation of the sum of all the z-scores into a new value that is then transformed into a new z-score (total or motor quotient z-score). That is the case for KTK3+ [12], or the TMC [11] (although the latter uses a general z-score value for each test, and not an age- and sex-related z-score). These methods are similar to the technique we used, where the mean and standard deviation used to find the z-scores results from a previous populational study. Other tests, such as the PDMS-2 [13], the TGMD-3 [14], or the MABC-2 [15], include a first transformation of the raw values into age- and sex-related standard scores (with a range of 1 to 10, or 1 to 20), and then its sum into a new standard score or percentile interval to represent the total motor quotient or percentile classification. In this case, each raw value is transformed two times, losing some information in each transformation, and resulting in less discriminating scores than the MCA.

A limitation of our results is the fact that no single gold standard is established for these specific measures of the MCA (sub-scales and total score). As explained previously, we choose the WAGM model as the most appropriate and complete model to determine the real value of the MCA scores. We also intend, in the future, to compare the MCA scores with other batteries used for assessing MC such as the KTK 3+, TMC, MABC-2, TGMD-3, or the Bruininks–Oseretsky.

## 5. Practical Implications

This study represents the final step for validating the MCA testing battery as a tool for assessing motor competence across all ages. MCA stability, locomotor, and manipulative sub-scales and MCA total score values can now be used when testing motor competence with any age and sex. The average of the normative age- and sex-related values for each test can be used to calculate the MCA sub-scales and total MCA scores. MCA is now ready to be used as a classification tool for MC throughout the human lifespan in different contexts (research, clinical, sports, education). Classification of the MCA cut-off values for the motor-impaired, underperformers, and high performers can now be established.

## 6. Conclusions

In conclusion, we advocate the use of the transformed percentile values to evaluate the performance on each test, and the average of these values to assess each sub-scale and the total MCA score, as the best and most effective method to score and classify MCA. With these insights, the MCA is ready to be used as a classification tool for MC throughout the lifespan in different contexts (research, clinical, sports, education). Further classification cut-off values for the motor-impaired, underperformers, and high performers can now be procured in future works.

## Figures and Tables

**Figure 1 children-09-01769-f001:**
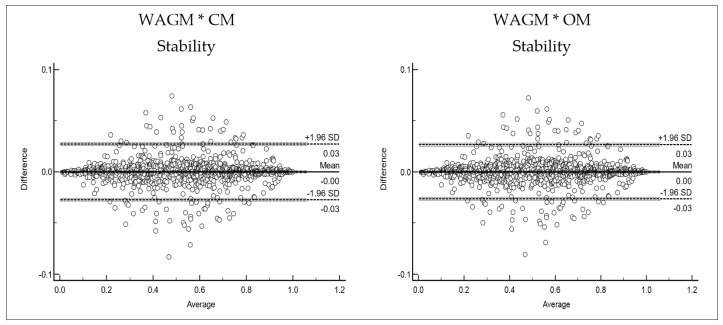
Bland–Altman graphs representing differences between the predicted models and the Weighted Age Group Model (WAGM) reference values for the MCA sub-scales, according to the percentile positions of the reference values of the WAGM vs. the mean of the two measurements, with limits of agreement and respective 95% confidence intervals.

**Table 1 children-09-01769-t001:** Loading coefficients for each sub-scale of the three tested models.

			Weighted Age Group Model				Overall Model	Configural Model
			3 to 6	7 to 10	11 to 16	17 to 22		
Stability		LJ	0.86	0.85	0.77	0.73	0.94	1
		SP	0.90	0.59	0.82	0.66	0.93	1
Locomotor		SLJ	0.89	0.63	0.94	0.88	0.94	1
		SHR	−0.82	−0.86	−0.86	−0.79	0.89	1
Manipulative		BKV	0.82	0.85	0.89	0.93	0.97	1
		BTV	0.78	0.76	0.91	0.91	0.95	1

LJs—lateral jumps; SPs—shifting platforms; SLJ—standing long jump; SHR—shuttle run; BKV. ball kicking velocity; BTV—ball throwing velocity.

**Table 2 children-09-01769-t002:** Reliability and Bland–Altman agreement analysis (mean bias and 95% limits of agreement) between the MCA sub-scale scores. Comparisons of the Configural Model and the Overall Model procedures with the Weighted Age Group Model.

	Cronbach’s Alpha	Intraclass Correlation Coefficient	Bland–Altman			
			Mean Bias (95%CI)		Lower Limit	Upper Limit
Stability						
WAGM * CM	0.999	0.999	0.00001 (−0.0009 to 0.0009)		−0.0271	0.0271
WAGM * OM	0.999	0.999	−0.00006 (−0.0009 to 0.0008)		−0.0266	0.0264
Locomotor						
WAGM * CM	0.999	0.999	−0.0006 (−0.0015 to 0.0001)		−0.0261	0.0248
WAGM * OM	0.999	0.999	0.000500 (−0.0004 to 0.0013)		−0.0263	0.0272
Manipulative						
WAGM * CM	1.000	1.000	−0.00007 (−0.0004 to 0.0002)		−0.0097	0.0096
WAGM * OM	1.000	1.000	−0.000006 (−0.0004 to 0.0004)		−0.012	0.012

WAGM—Weighted Age Group Model; CM—Configural Model; OM—Overall Model.

**Table 3 children-09-01769-t003:** Reliability and Bland–Altman agreement analysis (mean bias and 95% limits of agreement) between the MCA total scores). Comparisons of the Configural Model and the Overall Model procedures with the Weighted Age Group Model.

	Cronbach’s Alpha	Intraclass Correlation Coefficient	Bland–Altman			
			Mean Bias (95%CI)		Lower Limit	Upper Limit
Total MCA						
WAGM * CM	0.999	0.999	−0.0006458 (−0.0010 to 0.000004)		−0.01779	0.01649
WAGM * OM	1.000	1.000	−0.0005122 (−0.0011 to −0.0001)		−0.01681	0.01578

WAGM—Weighted Age Group Model; CM—Configural Model; OM—Overall Model

## Data Availability

Contact the correspondent author for data information.

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
