# Peer review of "Motor Competence Assessment (MCA) Scoring Method"

_children, 2022, doi:10.3390/children9111769_

Round 1
Reviewer 1 Report
Comments:
Introduction:
1. The TMC does indeed include a fine motor component, why is this a problem? Are the authors primarily concerned with gross motor competence? What is the difference between manipulative gross motor skills and manipulative (fine) motor skill? Please clarify in the introduction, what definitions that apply to the current study.
2. I feel that this paper is very hard to justify since there is no argumentation/justification/rationale in the introduction for the different computational approaches towards MCA scores. Why did the authors expect differences in the computational method? Without this information, the study feels de-contextualized and hard to interpret. Please provide in the introduction, at last a couple of paragraphs on the empirical and theoretical rationale behind the various statistical methods investigated.
Materials and methods
3. Although I understand that the MCA is thoroughly described in previous studies, the current paper would be easier to evaluate if at least some basic information with regard to test items and standardization is presented.
4. Again, the results are difficult to interpret since no argumentation/justification/rationale is provided for the Weighed Age-Group Model, the overall (?) model and the configural model. Why would these potentially classify children differently?
Discussion
5. The claim that the paper aims at deciding ‘on the better algorithm’ (which is indeed a relevant question), cannot be justified without providing at least some empirical/theoretical rationale for it to be evaluated against. That all generated roughly the same results does not, in principle, justify any of the computational methods.
6. I certainly agree that no single gold standard exist for computing MC scale/subscale, this further highlights the need for additional information to evaluate against.
7. At seems to me that the MCA battery includes many similar items compared to e.g., KTK, TMC and TGMD. The question therefore arises why are these other batteries are considered to be proxy measures for MC?
8. It seems to me that if the authors have ambitions for deciding upon cut-off values for potentially motor impaired children, the validation process is far from finalized!
Reviewer 2 Report
The authors want to provide a psychometric validation of the MCA. However, this is not a psychometric validation. They compare three models for assessing the score that should be obtained for the MCA.
The analyses are based on agreement and reliability analyses: authors try to determine whether the reliability between the three methods is high, which is the case. However, this is a truism. It is self-evident that the same individual scores used with three methods to obtain a total score will lead to total scores quite similar. I am afraid that this paper does not bring important knowledge to the scientific community, and I do not know how I could give some advice to authors to modify their manuscript to reach the minimum standard from the information given in the manuscript. At the best, it could have been a supplemental analysis in one of their previous papers. In my view, what the authors made is salami slicing, which belongs to the QRP.
Reviewer 3 Report
The paper looks very interesting.
The purpose of the study is not clear.
Avoid multiple paragraphs.
Methods:
First three lines are confusing.
Rewrite the content.
The setting
sample size calculations are not very clear.
Results :
Well explained.
Discussion:
Discussion on the need for the development and use of the tool.
Repetition of results can be avoided
Limitations have not been mentioned clearly.
conclusion based on the objective.
References were accurate
Round 2
Reviewer 1 Report
Congratulations, and thank you for taking my comments into consideration.
Author Response
Thanks for your contribution to our work.
Reviewer 2 Report
- first, it was clear that the paper did not bring new information to the scientific litterature
- second, authors used what is known as salami slicing which is a questionnable practice research.
Author Response
We agree on disagreeing.
Thanks for your comments.
Reviewer 3 Report
All the quires have been addressed in revised version.
still there are a couple issues should be addressed.
especially limitations and strengths of the study.
it is better to consider and accept limitations since the status has been done already.
Author Response
Thanks for your comments.
We believe all have been addressed in the manuscript.